# The Expression of Toll-like Receptors (TLR7 and TLR9) in Class III and Class IV of Recently Diagnosed Lupus Nephritis with 12-Month Follow-Up

**DOI:** 10.3390/ijms25137023

**Published:** 2024-06-27

**Authors:** José Ignacio Cerrillos-Gutiérrez, Miguel Medina-Pérez, Jorge Andrade-Sierra, Andrés García-Sánchez, Ernesto Germán Cardona-Muñoz, Wendy Campos-Pérez, Erika Martínez-López, Daniela Itzel Sánchez-Lozano, Tannia Isabel Campos-Bayardo, Daniel Román-Rojas, Luis Francisco Gómez-Hermosillo, Jorge Casillas-Moreno, Alejandra Guillermina Miranda-Díaz

**Affiliations:** 1Department of Nephrology, National Medical Center of the West, Mexican Social Security Institute, Guadalajara 44340, Jalisco, Mexico; chachisnefro@hotmail.com (J.I.C.-G.); enepis1@yahoo.com.mx (M.M.-P.); jorg_andrade@hotmail.com (J.A.-S.); 2Department of Physiology, University Center of Health Sciences, University of Guadalajara, Guadalajara 44360, Jalisco, Mexico; andres_garciasanchez_3@hotmail.com (A.G.-S.); cameg1@gmail.com (E.G.C.-M.); itzel.10274@gmail.com (D.I.S.-L.); tanniaisabelcb@gmail.com (T.I.C.-B.); daniel.rrojas@academicos.udg.mx (D.R.-R.); 3Department of Molecular Biology and Genomics, Institute of Nutrigenetics and Translational Nutrigenomics, University of Guadalajara, Guadalajara 44340, Jalisco, Mexico; wendy_yareni91@hotmail.com (W.C.-P.); erikamtz27@yahoo.com.mx (E.M.-L.); 4Department of Laparoscopic Surgery, Hospital Civil de Guadalajara, “Juan I Menchaca”, Guadalajara 44360, Jalisco, Mexico; luisgomez_53@hotmail.com (L.F.G.-H.); jcasillas_moreno@hotmail.com (J.C.-M.)

**Keywords:** systemic lupus erythematosus, lupus nephritis, innate immunity, acquired immunity, TLR7, TLR9

## Abstract

Renal involvement is an important cause of morbidity and mortality in systemic lupus erythematosus (SLE). The present study included patients with recently diagnosed Class III and Class IV lupus nephritis (LN) treated by Rheumatology who, upon the detection of alterations in their kidney function, were referred to Nephrology for the joint management of both medical specialties. The purpose of this study was to compare the plasma expression of Toll-Like Receptor 7 (TLR7) and TLR9 in healthy control (HC) subjects and newly diagnosed Class III and Class IV LN patients with 12-month follow-ups. The plasma expression of TLR7 and TLR9 proteins was determined by the ELISA method. A significant increase in the expression of TLR7 protein was found in Class III LN in the basal determination compared to the expression in the HC (*p* = 0.002) and at 12 months of follow-up (*p* = 0.03) vs. HC. The expression of TLR9 showed a behavior opposite to that of TLR7. TLR9 showed decreased protein expression in LN Class III patients’ baseline and final measurements. The result was similar in the basal and final determinations of LN Class IV compared to the expression in HC. A significant decrease in SLEDAI -2K was observed at 12 months of follow-up in patients in Class III (*p* = 0.01) and Class IV (*p* = 0.0001) of LN. Complement C3 levels improved significantly at 12-month follow-up in Class IV patients (*p* = 0.0001). Complement C4 levels decreased significantly at 12-month follow-up in LN Class III compared to baseline (*p* = 0.01). Anti-DNA antibodies decreased significantly at 12 months of follow-up in Class IV LN (*p* = 0.01). A significant increase in proteinuria was found at 12 months of follow-up in Class III LN, compared to the baseline determination (*p* = 0.02). In LN Class IV, proteinuria decreased at 12 months of follow-up compared to baseline (*p* = 0.0001). Albuminuria decreased at 12 months of follow-up in LN Class IV (*p* = 0.006). Class IV LN, albuminuria also decreased at 12 months of follow-up (*p* = 0.009). Hematuria persisted in all patients and the glomerular filtration rate did not change. Three Class IV patients died before 12 months of follow-up from various causes. In conclusion, although the rheumatologic data appeared to improve, the renal function data remained inconsistent. Decreased expression of TLR9 and increased expression of TLR7 could be useful in the early diagnosis of Class III and Class IV LN is correct.

## 1. Introduction

Systemic lupus erythematosus (SLE) is an autoimmune disease that is defined by the inadequate activation of self-reactive T and B cells and the production of autoantibodies and immune complexes (IC) capable of producing irreparable organ damage [1]. SLE is the prototype of an autoimmune disease characterized by the loss of tolerance to antinuclear antigens [2]. Approximately 40% of SLE patients have defects in the clearance of apoptotic cells that are commonly cleared rapidly by phagocytes in healthy individuals [3]. SLE presents with a wide range of clinical manifestations, including dermatological, neuropsychiatric, cardiovascular, and renal symptoms [4]. The incidence of SLE is 0.3–31.5 per 100,000 inhabitants per year and the adjusted prevalence approaches or even exceeds 50–100 per 100,000 inhabitants [5]. For effective treatment, it is critical to improve our ability to diagnose SLE early.

Toll-like receptors (TLRs) are a family of evolutionarily conserved innate immune receptors that play a crucial role in first-line defense against foreign molecules called pathogen-associated molecular patterns (PAMPs) [4]. There are intracellular receptors that are expressed in the endosomal compartments of cells. These receptors detect nucleic acids such as single-stranded RNA (ssRNA) in TLR7 and TLR8 or unmethylated single-stranded DNA containing cytosine-phosphate-guanine (CpG) motifs in TLR9 is common in viral infections and bacterial genomes [5]. TLRs recognize and respond to endogenous ligands produced during an inflammatory state or tissue damage caused by systemic autoimmune diseases through the production of specific autoantibodies [6]. Self-antigens are derived from chromatin and small nuclear ribonucleoproteins (snRNPs), which are normally sequestered from the immune system by their intracellular location [7]. Nuclear antigens become more accessible to TLRs as a result of cell death or apoptosis. When nuclear antigens are activated, TLRs rapidly act through adapter proteins to induce transcription factors for type I interferon (IFN) and other pro-inflammatory mediators that contribute to the development and progression of autoimmune disease. The identification and characterization of endogenous TLR ligands offer a novel perspective to explore the etiology of autoimmune diseases [8]. TLR signaling also promotes three key activities through which B cells may contribute to autoimmune disease: the production of antibodies, the presentation of antigens to T cells, and the production of cytokines. The above suggests the important roles of B cell signaling and TLRs in autoimmune diseases [9]. The genetic association has been implicated in TLR signaling in SLE [10]. TLR7 and TLR9 are intracellular receptors not directly accessible to natural extracellular nucleic acids such as B cells because they do not internalize extracellular material through micropinocytosis or endocytosis [11]. TLR7 expression is higher in women than in men due to the location of TLR7 on the X chromosome. Normally, one X chromosome is inactivated in women. However, some genes are upregulated and more effective in women than men’s cells [12]. Reducing TLR7 activity could slow the development of SLE [13]. TLR3, TLR7, and TLR9 were reported in the kidneys of patients with lupus nephritis (LN) in correlation with the clinic-pathological indices [14]. TLR9 expression is positively regulated in the interstitium tubule of patients with LN [15]. There is limited information on the plasma expression of TLR7 and TLR9 in patients with recently diagnosed Class III and Class IV LN and its evolution over time.

The purpose of this study was to compare the plasma expression of TLR7 and TLR9 in HC and in recently diagnosed Class III and Class IV lupus nephritis (LN) patients with 12-month follow-up. 

## 2. Results

In this study, 31 patients with a recent diagnosis of LN were included, in which 15 corresponded to Class III of LN, 16 patients to Class IV of LN, and 17 subjects formed the HC group. In the HC group, 7 men and 10 women were included with an age of 34.67 ± 3.18 years, a height of 166 ± 0.01 cm, and a body weight of 74.57 ± 6.40 Kg. In the Class III of LN group, there were t3 men (20%) and 12 women (80%). Three men (18.75%) and 13 women (81.25%) were included in Class IV LN group. Before completing the 12 months of follow-up, three patients in Class IV LN group died. A female patient died from hypovolemic shock by uncontrollable digestive bleeding, another female patient died from COVID-19, and the male died from acute lung edema. The SLE patients were treated with glucocorticoids, non-steroidal anti-inflammatory drugs, antimalarials, immunosuppressants, and biological products by the Department of Rheumatology.

The average age of patients in the LN Class III group was 28.12 ± 1.75 years, and in the LN Class IV group, it was 31.56 ± 1.82 years. The body weight was similar in patients in the LN Class III group, being 67.35 ± 5.47 K, and in the LN Class IV group, being 63.97 ± 3.66 K. The BMI was also similar in both LN Classes. The significant biochemical data were the increases in platelets at 12 months of follow-up in LN Class IV patients (*p* = 0.05). A decrease in chloride was found at 12 months of follow-up in the LN Class III patients (*p* = 0.05). The significant decrease in SLEDAI-2K at 12 months of follow-up in the Class III and Class IV of LN groups is striking. In LN Class III patients, the baseline SLEDAI-2K was 18.53 ± 1.87 and 9.88 ± 2.61 at 12-month follow-up (*p* = 0.01). The baseline SLEDAI-2K in the LN Class IV patients was 23.31 ± 1.63 and 10.80 ± 1.74 at 12-month follow-up. (*p* = 0.0001). The complement C3 increased significantly in LN Class IV patients at 12 months of follow-up (*p* = 0.0001). Complement C4 decreased significantly at 12 months of follow-up in LN Class III patients (*p* = 0.01). Anti-DNA antibodies decreased in LN Class IV patients at 12 months of follow-up (*p* = 0.01) and became negative at 12 months of follow-up in LN Class III patients. The significant increase in proteinuria in LN Class III patients at 12 months of follow-up is striking (*p* = 0.02). Proteinuria decreased in LN Class IV patients at 12 months of follow-up (*p* = 0.0001). Albuminuria decreased significantly at 12 months of follow-up in LN Class III patients from 74.04 ± 24.37 mg/24 h to 1.88 ± 0.00 mg/24 h (*p* = 0.006). Albuminuria decreased significantly from 111.23 ± 29.60 mg/24 h at baseline in LN Class IV patients to 19.29 ± 7.15 mg/24 h at 12-month follow-up (*p* = 0.009). The hematuria and glomerular filtration rates at baseline and 12 months of follow-up in both LN Class III and Class IV patients did not change during the duration of the study (Table 1).

Table 1 A significant decrease in chloride was observed at 12 months of follow-up in Class III LN patients as well as an increase in platelets at 12 months of follow-up in Class IV LN. SLEDAI-2K and anti-DNA antibodies decreased significantly at 12 months of follow-up in Class III and Class IV LN patients. C3 increased significantly at 12-month follow-up in LN Class IV patients and C4 decreased in LN Class III patients. Albuminuria decreased at 12 months of follow-up in LN Class III and Class IV patients, as did proteinuria in LN Class IV patients. Proteinuria increased at 12 months of follow-up in LN Class III patients.

Table 2 and Figure 1 show the overexpression of TLR7 in the baseline determination, which was 2.44 ± 0.65 ng/mL for LN Class III patients compared to the levels obtained in HC, 0.41 ± 0.10 ng/mL (*p* = 0.002). The overexpression of TLR7 persisted at 12 months of follow-up, being 1.05 ± 0.30 ng/mL vs. the expression of HC (*p* = 0.03). The baseline TLR7 expression of the LN Class IV patients, 0.45 ± 0.11 ng/mL, was similar to the expression found in HC, 0.41 ± 0.17 ng/mL, and at 12 months of follow-up, 0.83 ± 0.24 ng/mL (*p* = 0.83).

The TLR9 expression found in HCs was higher, 1.28 ± 0.45 ng/mL, compared to baseline levels, 0.56 ± 0.15 ng/mL, and 12 months of follow-up, 0.52 ± 0.09 ng/mL, in LN Class III patients. The same phenomenon was observed in LN Class IV patients, where the baseline levels, 0.52 ± 0.09 ng/mL, the levels at 12 months of follow-up, 0.56 ± 0.06 ng/mL, and the expression in the HC, 1.28 ± 0.45 ng/mL, did not show significant differences. 

A cross-sectional correlation was performed to determine the association between clinical parameters related to renal function and rheumatological data at baseline. A positive correlation was found between TLR7 and CRP values for NL class IV patients (Table 3).

Table 2 The overexpression of TLR7 can be observed in the baseline determination and at 12 months of follow-up in LN Class III patients compared to the expression found in HC. The expression of TLR9 was higher in HC compared to the expression in the baseline determination and at 12 months of follow-up of LN Class III and Class IV patients.

Table 3 Cross-sectional correlation between basal TRL7 and TRL9 with clinical parameters of renal function and rheumatological data. Baseline TLR7 values have a positive correlation for patients with NL class IV. Other clinical variables related to kidney function and rheumatological data did not show a correlation with TLR7 and TLR9.

## 3. Discussion

In the present study, we address the protein expression of TLR7 and TLR9 determined by ELISA methods, in addition to the modifications found in the renal and rheumatological functions of recently diagnosed Class III and Class IV LN patients at 12 months of follow-up. 

The overexpression found in the TLR7 protein was notable in the baseline determination and at 12 months of follow-up in Class III LN patients compared to the expression in HC, without this finding being reflected in the Class IV LN patients. An important background of our findings is what was reported in 2019 in murine models of SLE and in patients, where the authors emphasize the participation of the TLR7 protein in the onset and progression of the disease. It was even revealed that the expression of TLR7 in immune cells depended on the TLR7 transgene itself and/or on factors specific to each cell in SLE [16]. TLR7 is essential for the activation of RNA-associated IC because TLR7 recognizes ssRNA essential for host defense against the development and progression of diseases caused by autoimmunity such as SLE [17,18]. Increased TLR7 protein expression was recently reported in renal macrophages and dendritic cells (DCs). The authors demonstrated the affectation of the frequency of leukocyte infiltration in the kidney. The authors suggest controlling TLR7 levels within myeloid populations to prevent chronic inflammation and severe nephritis in patients predisposed to SLE [19]. TLR7 drives the extra-follicular B cell response and germinal center reaction involving autoantibody production and disease pathogenesis [20]. Taking into consideration what was previously reported, it must be considered that perhaps the inflammatory state of the Class III LN patients was uncontrolled, which could favor the aggravation of LN by favoring the destruction of tissues, with the possibility of triggering organic insufficiency [21]. In contrast, we consider that finding similar levels of TLR7 in HC in LN Class IV patients may not be conclusive; more research needs to be done with a larger number of patients. On the other hand, we did not investigate any TLR polymorphism in LN patients in the present study. Some polymorphisms favor greater expression of TLR7 ssRNA ligands that are associated with a higher risk of developing SLE [22]. An important factor in the present study and in what has been published in the scientific literature is the higher expression of TLR7 in women than in men due to the location of TLR7 on the X chromosome [23]. This observation is consistent with the higher prevalence of SLE in women than in men [21]. In the present study, women predominate in both classes of LN. However, it is worth searching soon for some polymorphisms capable of influencing the progression of SLE.

On the other hand, we found a non-significant decrease in the plasma expression of TLR9 in LN Class III and Class IV patients at baseline and 12 months of follow-up compared to the expression found in HC. We consider this finding to be important because different functions of TLR9 have been identified in different lupus-like disease models. In some models, TLR9 exerted protective effects. In 2015, it was reported that mice with null TLR9 showed more severe lupus than control mice in which greater immunoglobulin deposits and more severe LN were found. Disease exacerbation was abrogated when TLR7 was also deleted from TLR9 null mice, indicating that TLR9 could limit lupus pathogenesis by diminishing the deleterious effects of TLR7. TLR9 restricts TLR7 activity in DCs and B cells, respectively [24]. The decrease in TLR9 expression in patients with Class III and Class IV LN compared to the higher expression found in HC is consistent with the previously reported in the literature, which suggests an increase in the pathogenesis of LN because the activity of TLR7 is not restricted. 

Although SLEDAI-2K significantly decreased at 12-month follow-up in patients from both LN classes, the result was not reflected in the C3 of LN Class III patients. The complement C4 declined significantly in LN Class III patients at 12 months of follow-up, suggesting worsening of the disease. We emphasize that complement plays a vital role in the pathogenesis of SLE. In general, complement levels in blood and complement deposition in histological tests are used to evaluate the treatment of SLE [25]. Determining the complement status is useful in diagnosis, monitoring disease activity, response to treatment, and predicting disease prognosis. The results obtained after 12 months of follow-up were inconsistent and worrying because they could translate into inadequate response to the treatment administered by the treating physicians (Rheumatologist and Nephrologist).

The renal function results were also inconsistent and worrying because there was an increase in proteinuria in LN Class III patients at 12 months of follow-up and a decrease in LN Class IV patients in contrast to the decrease in albuminuria. Hematuria did not disappear and the glomerular filtration rate did not improve at the end of the follow-up. It must be considered that the remission or stabilization of LN was not achieved 12 months after consistently administered treatment. Three patients in Class IV even died before completing follow-up. Failure to achieve the remission of LN is reported in the available scientific literature as an independent prognostic factor for mortality [14,26]. 

The causes of death of the three Class IV LN patients that occurred before 12 months of follow-up were diverse and consistent with what was recently published, where the authors carried out a population study and demonstrated a persistent mortality gap for all causes and specific causes in patients with SLE compared to the general population [27,28], as reported in the present study.

## 4. Materials and Methods

A prospective cohort study was carried out with a twelve-month follow-up. Patients who agreed to participate in the study included women and men over 18 years of age with SLE, with a recent diagnosis of focal LN (Class III) or diffuse LN (Class IV). The patients were selected from the Nephrology Department of the Specialty Hospital of the National Medical Center of the West of the Mexican Institute of Social Security (IMSS) in Guadalajara, Jalisco, Mexico. When the treating rheumatologist found alterations in kidney function, they sent the patient to the Nephrology Department where a blood sample and kidney biopsy were performed. According to the histopathology result, the patients were classified into Class III and Class IV of LN. All patients were treated by both specialists (rheumatologist and nephrologist). Patients with data of any infectious or neoplastic process were not included. Patients with any type of diabetes, a history of other autoimmune diseases, or thrombotic events were excluded. Patients who withdrew informed consent before 12 months of follow-up were also excluded. Below is the LN project flowchart (Figure 2).

Data recorded for analysis were gender, age, height, and body weight. Biochemical data included hemoglobin, hematocrit, platelets, glucose, albumin, chloride, potassium, phosphorus, calcium, sodium, and magnesium. Renal function data included glomerular filtration rate, urea, creatinine, proteinuria, albuminuria, and hematuria. SLE disease activity was determined using the index (SLEDAI-2K) [29] C3, C4, and anti-DNA antibodies. The diagnosis of LN Class was determined by renal biopsy within the first three months of the detection of LN according to the criteria of the WHO or the International Society of Nephrology and the Renal Pathology Society (ISN)/RPS) of 2003 [30]. Patients with Class III (focal) LN are characterized by proteinuria, hematuria, increased serum creatinine, and sometimes nephrotic syndrome and arterial hypertension. Class IV (diffuse) is characterized by the progression of the lesion depending on the percentage of glomeruli affected [29]. Venous blood samples (5 mL in a dry tube and 5 mL in a tube with 7.2 mg of potassium ethylenediaminetetraacetic acid) were collected at the first visit to the Nephrology Department and at 12 months of follow-up. The samples were centrifuged at 3500 rpm for 10 min to obtain plasma. Aliquots were stored at −80 °C for later analysis. 10 mL of extra blood was obtained from 17 blood donors who came to the blood bank and who agreed to form the healthy control group (HC). These samples were used to establish the normal values of the hTLR7 and hTLR9 reagents. 

### 4.1. hTLR7 y hTLR9

For the processing of the reagents, the manufacturer’s instructions for TLR7 and TLR9 were followed. The kit was a sandwich enzyme-linked immunosorbent assay (ELISA) designed for the in vitro quantitative determination of human TLR7 (A78893) and TLR9 (A2423) in plasma (https://www.antibodies.com/products/search=tlr9%20(a2423)%20, accessed on 16 May 2024, Cambridge, UK). Amounts of 100 µL of the dilutions of the standard and samples were added to the wells, and the plate was covered and incubated at 37 °C for 2 h. The liquid was removed from the wells and, without washing the plate, 100 µL of the detection reagent was added. The plate was incubated for 1 h at 37 °C. The washing solution was aspirated and the plate was allowed to rest for 2 min 3 times. A total of 100 µL of the detection antibody was added and the plate was incubated for 1 h at 37 °C. After washing the plate 5 times, 90 µL of the substrate solution was added and was incubated for 20 min at 37 °C. Subsequently, 50 µL of the stop solution was added to each well and the plate was read at 450 nm optical density to obtain the absorbance of the reagents and obtain the results according to the standard curve.

### 4.2. Ethical Considerations

This study was conducted according to the ethical principles for medical research on human subjects stipulated in the Declaration of Helsinki 64th General Assembly, Fortaleza Brazil, October 2013, the Standards of Good Clinical Practice according to the guidelines of the International Conference on Harmonization. To safeguard the confidentiality of patient data, codes were assigned to the samples in accordance with the provisions of the General Health Law of Mexico, and according to the Regulations of the General Health Law on Research for Health, art. 17, which corresponds to a category II study. All patients signed the informed consent form in the presence of witnesses. The study was approved by the local Ethics and Research Committees (IMSS) on 12 January 2022, with the registration number: R-2022-1301-010.

### 4.3. Statistical Analysis

Normally distributed data were presented as mean ± error standard (ES). Categorical variables were expressed as frequency and percentage. According to the type of data distribution, all demographic and related characteristics were compared between baseline and twelve-month follow-up determination using Chi^2^. Shapiro–Wilk test was used to determine the distribution of values. For parametric comparisons, paired-sample *t*-test and independent sample *t*-test were used. For non-parameter comparisons, Wilcoxon and Mann–Whitney U tests were used, as well as paired-sampled *t*-tests and independent sample *t*-tests. Statistical analysis was performed using IBM SPSS v.18 software (Chicago, IL, USA). Any value of *p* < 0.05 was considered significant. 

## 5. Conclusions

LN represents the most serious organic manifestation of SLE in terms of morbidity and mortality. TLR modulation and signaling have become important strategies for the diagnosis and treatment of SLE. In the present study, we found a significant imbalance in TLR7 expression characterized by the increases seen in Class III and Class IV patients at baseline and 12 months of follow-up compared to the expression found in HC. In contrast, TLR9 expression was found to be decreased in newly diagnosed Class III and Class IV LN patients at baseline and at 12-month follow-up vs. the expression in HC. Determining low levels of TLR9 and increased TLR7 could be useful in the early diagnosis of patients with LN.

Although there was an apparent improvement in the rheumatologic data of SLEDAI-2K, C3, C4, and anti-DNA antibodies, the renal function was inconsistent with no improvement. Therefore, more studies are required to establish the role of TLR7 and TLR9 in mediating SLE disease progression.

Limitations of the study. A limited number of patients from Class III and Class IV of LN and short follow-up time were included. The selected sample belongs to a population that receives public health care. Medical care can delay the appearance of relevant clinical manifestations, so a cohort study would not be the best option to detect complications that are very rare. It is worth comparing the expression of both TLRs in LN with or without stable LN, with a larger number of patients and with a longer follow-up over time.

Strengths of the study. This work offers unusual information on the evolution of LN accompanied by the monitoring of immune markers that are little-known in their role in LN disease (TLR7 and TLR9). The 12-month prospective follow-up allowed us to identify the appearance of significant changes in clinical parameters based on the LN classification (Class III and Class IV).

## Figures and Tables

**Figure 1 ijms-25-07023-f001:**
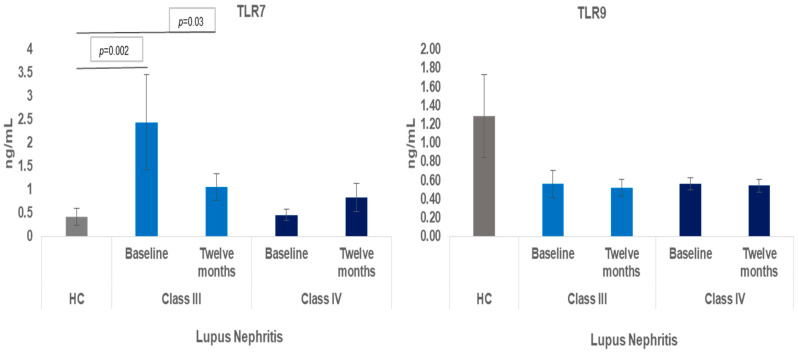
Expression of TLR7 and TLR9 in ln class III and class IV patients. Overexpression of TLR7 can be observed in the baseline determination and at 12 months of follow-up in LN Class III patients. The expression of TLR7 in Class IV LN was similar to that found in HC. The expression of TLR9 in the HC was higher than the expression found in Class III and Class IV of LN at baseline and at 12 months of follow-up.

**Figure 2 ijms-25-07023-f002:**
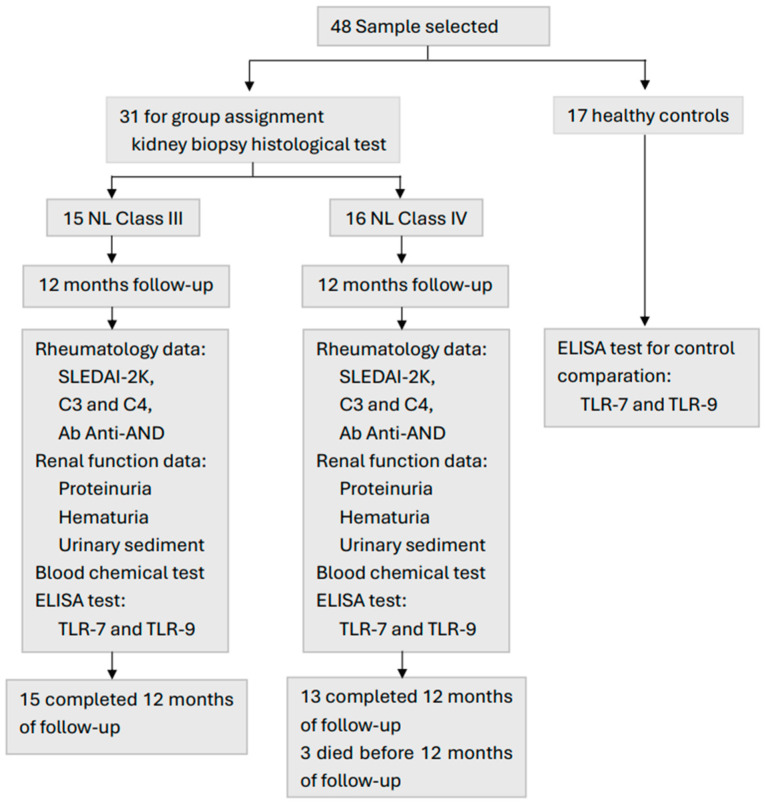
Data collection.

**Table 1 ijms-25-07023-t001:** Class III and Class IV Lupus Nephritis; Baseline vs. 12 Months Follow-up.

	Class III N = 15			Class IV N = 16		
	Baseline	Twelve Months	*p*	Baseline	Twelve Months	*p*
Gender N (%)						
Male	3 (20)			3 (18.75)		0.93 ^t^
Female	12 (80)			13 (81.25)		
Age years	28.12 ± 1.75			31.56 ± 1.82		0.18 ^t^
Body weight Kg	67.35 ± 5.47			63.97 ± 3.66		0.61 ^t^
Body mass index kg/m^2^	24.95 ± 1.26			25.66 ± 1.47		0.72 ^t^
Treatment						
Rituximab	4 (27)			4 (25)		0.92
Tacrolimus	0 (0)			2 (13)		0.16
Cyclosporine	6 (40)			1 (6)		0.025 ^t^
Mycophenolate mofetil	11 (73)			10 (62)		0.89
Methylprednisolone	9 (60)			10 (62)		0.52
Prednisone	11 (73)			14 (87)		0.32
Azathioprine	1 (7)			1 (6)		
Chloroquine	7 (47)			10 (62)		0.17
Biochemical data						
Hemoglobin g/dL	12.15 ± 0.57	12.37 ± 0.76	0.82	11.46 ± 0.56	12.47 ± 0.89	0.33
Hematocrit %	35.57 ± 1.52	37.72 ± 2.66	0.49	33.99 ± 1.56	36.85 ± 1.54	0.21
Platelets thousands/µL	239.33 ± 19.22	276.45 ± 19.68	0.19	231.44 ± 22.84	293.73 ± 19.82	0.05
Glucose mg/dL	89.93 ± 2.52	84.89 ± 3.66	0.27	96.64 ± 3.00	91.08 ± 4.05	0.27
Albumin g/dL	3.58 ± 0.23	3.46 ± 0.27	0.74	3.37 ± 0.17	3.46 ± 0.27	0.77
Chlorine mmol/L	109.75 ± 1.60	104.43 ± 2.14	0.05	107.35 ± 1.12	109.18 ± 3.49	0.60
Potassium mmol/L	4.35 ± 0.24	4.26 ± 0.25	0.80	4.54 ± 0.21	4.43 ± 0.14	0.68
Calcium mmol/L	3.09 ± 0.58	4.72 ± 0.89	0.14	2.93 ± 0.47	4.74 ± 1.02	0.10
Magnesium mmol/L	0.92 ± 0.10	1.03 ± 0.18	0.60	0.92 ± 0.07	1.29 ± 0.20	0.08
Sodium mmol/L	137.47 ± 0.76	136.96 ± 0.70	0.76	138.15 ± 0.82	139.09 ± 0.78	0.60
Rheumatological Data						
SLEDAI-2K	18.53 ± 1.87	9.88 ± 2.61	0.01	23.31 ± 1.63	10.80 ± 1.74	0.0001
C3 mg/dL	107.51 ± 9.84	91.78 ± 12.75	0.13	65.01 ± 4.50	104.68 ± 6.85	0.0001
C4 mg/dL	23.19 ± 1.87	14.98 ± 2.42	0.01	19.52 ± 3.57	20.45 ± 2.64	0.84
CRP mg/L	7.60 ± 1.64	6.49 ± 2.24	0.69	14.64 ± 1.63	9.83 ± 2.50	0.11
Ab anti-DNA IU/mL	111.49 ± 58.71	(-)	0.07	72.33 ± 17.04	22.36 ± 4.51	0.01
Renal Function						
Urea mg/dL	34.73 ± 2.22	50.43 ± 12.61	0.23	47.23 ± 5.31	46.21 ± 11.95	0.94
Creatinine mmol/L	1.11 ± 0.11	1.89 ± 0.78	0.33	1.09 ± 0.17	1.65 ± 0.74	0.44
Phosphorus mmol/L	4.11 ± 0.22	4.00 ± 0.42	0.81	3.82 ± 0.22	3.92 ± 0.45	0.84
Proteinuria g/L	2.07 ± 0.40	28.33 ± 10.35	0.02	3.42 ± 0.44	0.76 ± 0.31	0.0001
Albuminuria mg/24	74.04 ± 24.37	1.88 ± 0.00	0.006	111.23 ± 29.60	19.29 ± 7.15	0.009
Urinary creatinine mg/kg/day	2.99 ± 0.84	31.40 ± 15.23	0.07	2.04 ± 0.54	1.65 ± 0.22	0.53
Hematuria erythrocytes	40.54 ± 18.61	10.80 ± 0.00	0.12	35.00 ± 12.92	31.43 ± 20.80	0.88
Glomerular filtration rate mL/min/1.73 m^2^	80.60 ± 7.36	93.49 ± 13.88	0.42	92.84 ± 10.60	87.65 ± 14.20	0.77

Values are expressed as mean ± standard error (SE). Student’s *t*-test of dependent samples and Wilcoxon test were used. ^t^ Class III vs. Class IV baseline comparation. Chi^2^ was used for dichotomous variables. CRP = C-reactive protein, C3 = complement protein C3, C4 = complement protein C4.

**Table 2 ijms-25-07023-t002:** TLR7 and TLR9 in Class III and Class IV Lupus Nephritis vs. Healthy Control (HC).

	HC		Class III				Class IV		
	N = 17	Baseline N = 15	*p* HC vs. Baseline	Twelve Months N = 15	*p* Baseline vs. Twelve Months	Baseline N = 16	*p* Baseline vs. HC	Twelve Months N = 13	*p* Baseline vs. Twelve Month
TLR7 ng/mL	0.41 ± 0.17	2.44 ± 1.01	0.002	1.05 ± 0.28	0.03	0.45 ± 0.12	0.79	0.83 ± 0.30	0.08
TLR9 ng/mL	1.28 ± 0.45	0.56 ± 0.15	0.17	0.52 ± 0.09	0.17	0.56 ± 0.06	0.13	0.54 ± 0.07	0.18

Values are expressed as mean ± standard error (SED). Wilcoxon and Mann–Whitney U tests were used.

**Table 3 ijms-25-07023-t003:** Cross-sectional correlation between basal TLR7 and TLR9 with clinical parameters of renal function and rheumatological data.

	Class III N = 15				Class IV N = 16			
	TLR7		TLR9		TLR7		TLR9	
	Rho	*p*	Rho	*p*	Rho	*p*	Rho	*p*
SLEDAI-2K	0.116	0.71	−0.061	0.84	0.171	0.53	0.317	0.23
C3	0.104	0.73	−0.187	0.54	−0.395	0.13	0.110	0.68
C4	−0.005	0.99	−0.300	0.32	−0.469	0.07	−0.147	0.59
CRP	−0.281	0.35	0.209	0.49	0.540	0.031	−0.409	0.12
Urea	0.093	0.76	0.319	0.29	0.053	0.85	0.077	0.79
Creatinine	−0.011	0.97	−0.319	0.29	−0.145	0.59	0.024	0.93
GFR	−0.069	0.82	0.354	0.24	−0.140	0.62	0.159	0.57
Proteinuria	0.005	0.99	0.374	0.21	0.115	0.67	−0.056	0.84
Urinary creatinine	−0.295	0.33	0.148	0.63	−0.044	0.87	0.141	0.60
Hematuria	−0.173	0.61	−0.290	0.39	−0.189	0.56	0.315	0.32
Phosphorus	−0.180	0.58	0.322	0.31	0.178	0.51	−0.160	0.55
Calcium	0.198	0.52	0.281	0.35	0.028	0.92	−0.130	0.63
Magnesium	−0.274	0.37	0.238	0.43	−0.137	0.61	0.189	0.48

Spearman’s Rho correlation test. GFR = Glomerular filtration rate, CRP = C—reactive protein, C3 = complement protein C3, C4 = complement protein C4.

## Data Availability

The database that supports the conclusions of this research work will be made available by the authors, upon express request and with the authorization of the Ethics and Research Committee.

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
