# Peer review of "The Expression of Toll-like Receptors (TLR7 and TLR9) in Class III and Class IV of Recently Diagnosed Lupus Nephritis with 12-Month Follow-Up"

_ijms, 2024, doi:10.3390/ijms25137023_

Round 1

Reviewer 1 Report

Comments and Suggestions for Authors

The authors compared TLR7 and TLR9 between patients diagnosed with LN class III and IV and healthy controls. In addition, the authors evaluated longitudinal changes in TLR7 and TLR9 levels. The focus points were very interesting, but the analysis at the current stage is insufficient considering the potential of the data. I would like to suggest further improvements based on the reviewers' comments.

Regarding the abstract:

What is the purpose and conclusion of this study?

Please state these clearly in the abstract. It is necessary.

There are several spelling mistakes in the abstract, such as NL (LN?) and TRL (TLR?), please correct them. Also, please spell out LN. I think it is Lupus Nephritis.

About the introduction:

What is the purpose of this study? Did the authors just want to compare TLR7 and TLR9 expression between LN III/IV and healthy controls? The purpose of this comparison is important.

Please check that the commas and full stops are placed correctly.

About the results:

For Table 1, what do "N-13" and "4 of 12" mean?

The quality of the tables is not good. Please update them. For example, the authors should change "N-15" to "N=15". At least it would be better to use brackets for the units of each parameter. If you want to emphasize "P<0.05" in bold, please be consistent in all tables.

Longitudinal data for TLR7 and TLR9 are valuable.

How about comparing the longitudinal changes in TLR7 and TLR9 between groups? How about comparing ΔTLR7 and 9 or using a linear mixed-effect model?

Is the title “Graph 1” appropriate for the authors’ instruction? I think “Figure” is better and more general.

The analyses of the association (or correlation) between TLR7 (and 9) and clinical characteristics should be informative. What kind of parameters are associated with TLR7 or TLR9?

The authors must have the details of pathological findings (because all patients were categorized into LN Classes). So, please analyze the association between pathological findings and TLR7 (and TLR9).

The authors defined a significant difference as p≤ 0.05, but p<0.05 is more common.

Are the healthy controls in this study appropriate for the study?

For example, did the authors compare between SLE patients without LN and patients with LN class III and IV? And/or between LN patients who achieved a complete remission and patients with LN class III and IV? This type of comparison should be helpful in drawing conclusions.

For the flow chart, please state how many patients were enrolled and then how many were excluded for specific reasons. Also, "N-15" should be changed to, for example, "N=15".

How did the authors select the healthy controls? Why was the n number of healthy controls 17?

For the methods, what is the “baseline”? please provide the definition.

For the data analysis of TLR7 and TLR9, were these values measured at the same plate (kit) on the same date? How does the distribution of the value of TLR7 and TLR9? Are these normal distributions without any standardization? I do not mean to criticize but I think these adjustments or quality control of the data may help the authors to find important results.

If the authors want to mention the data at 24 months, information on treatment should be important. Please add these to the results.

So please rethink the content and re-analyzIJe the data that the authors have.

Then I think the authors can come up with a better conclusion and state the purpose of the study. (For example, low levels of TLR7 and TLR9 may be useful to diagnose LN class III or IV). This kind of conclusion should be useful for clinicians.

Author Response

Reviewer 1 Document TLR7 and TLR9

Comments and Suggestions for Authors

The authors compared TLR7 and TLR9 between patients diagnosed with LN class III and IV and healthy controls. In addition, the authors evaluated longitudinal changes in TLR7 and TLR9 levels. The focus points were very interesting, but the analysis at the current stage is insufficient considering the potential of the data. I would like to suggest further improvements based on the reviewers' comments.

 Regarding the abstract:

Cuestion. What is the purpose and conclusion of this study? Please state these clearly in the abstract. It is necessary.

Answer. You right. The purpose of the study was to compare the plasma expression of TLR7 and TLR9 in HC and in recently diagnosed Class III and Class IV LN patients with twelve-month follow-up.

The conclusion of the study. Low expression of TLR9 and increased TLR7 could be useful in the recent diagnosis of Class III and Class IV LN.

It is worth comparing the expression of both TLRs in LN with or without stable LN; with a larger number of patients and with a longer follow-up over time.

 Comment. There are several spelling mistakes in the abstract, such as NL (LN?) and TRL (TLR?), please correct them. Also, please spell out LN. I think it is Lupus Nephritis.

Answer. I apologize for the careless error. I hope all grammatical and spelling errors have been corrected

About the introduction:

Comment. What is the purpose of this study? Did the authors just want to compare TLR7 and TLR9 expression between LN III/IV and healthy controls? The purpose of this comparison is important.

Answer. You are right, the objective of the study was confused, it was modified as follows

 The purpose of the study was to compare the plasma expression of TLR7 and TLR9 in HC and in recently diagnosed Class III and Class IV LN patients with twelve-month follow-up.

Comment. Please check that the commas and full stops are placed correctly.

Answer. The entire document was checked for grammar and spelling. I hope all the errors have been corrected

About the results:

Comment. For Table 1, what do "N-13" and "4 of 12" mean? The quality of the tables is not good. Please update them. For example, the authors should change "N-15" to "N=15". At least it would be better to use brackets for the units of each parameter. If you want to emphasize "P<0.05" in bold, please be consistent in all tables.

Answer. Thanks for the clarification. N-13 was removed. N=15 and N=16 were modified Table 1 was modified. We hope it is easier to understand

Table 1. Class III and  Class IV Lupus Nephritis; Baseline vs. Twelve Months Follow-up

Class III

N=15

Class IV

N=16

Baseline

Twelve months

p

Baseline

Twelve months

p

  Gender n (%)

     Male

3 (20)

3 (18.75)

0.93t

     Female

12 (80)

13 (81.25)

  Age years

28.12±1.75

31.56±1.82

0.18 t

  Body weight Kg

67.35±5.47

63.97±3.66

0.61 t

  Body mass index  kg/m2

24.95±1.26

25.66±1.47

0.72 t

Biochemical data

  Hemoglobin g/dL

12.15±0.57

12.37±0.76

0.82

11.46±0.56

12.47±0.89

0.33

  Hematocrit %

35.57±1.52

37.72±2.66

0.49

33.99±1.56

36.85±1.54

0.21

  Platelets thousands/µL

239.33±19.22

276.45±19.68

0.19

231.44±22.84

293.73±19.82

0.05

  Glucose mg/dL

89.93±2.52

84.89±3.66

0.27

96.64±3.00

91.08±4.05

0.27

  Albumin g/dL

3.58±0.23

3.46±0.27

0.74

3.37±0.17

3.46±0.27

0.77

  Chlorine mmol/L

109.75±1.60

104.43±2.14

0.05

107.35±1.12

109.18±3.49

0.60

  Potassium mmol/L

4.35±0.24

4.26±0.25

0.80

4.54±0.21

4.43±0.14

0.68

  Calcium mmol/L

3.09±0.58

4.72±0.89

0.14

2.93±0.47

4.74±1.02

0.10

  Magnesium mmol/L

0.92±0.10

1.03±0.18

0.60

0.92±0.07

1.29±0.20

0.08

  Sodium mmol/L

137.47±0.76

136.96±0.70

0.76

138.15±0.82

139.09±0.78

0.60

Rheumatological Data

  SLEDAI -2K

18.53±1.87

9.88±2.61

0.01

23.31±1.63

10.80±1.74

0.0001

  C3 mg/dL

107.51±9.84

91.78±12.75

0.13

65.01±4.50

104.68±6.85

0.0001

  C4 mg/dL

23.19±1.87

14.98±2.42

0.01

19.52±3.57

20.45±2.64

0.84

  CRP mg/L

7.60±1.64

6.49±2.24

0.69

14.64±1.63

9.83±2.50

0.11

  Ab anti-DNA IU/mL

111.49±58.71

( - )

0.07

72.33±17.04

22.36±4.51

0.01

Renal Function

  Urea mg/dL

34.73±2.22

50.43±12.61

0.23

47.23±5.31

46.21±11.95

0.94

  Creatinine mmol/L

1.11±0.11

1.89±0.78

0.33

1.09±0.17

1.65±0.74

0.44

  Phosphorus mmol/L

4.11±0.22

4.00±0.42

0.81

3.82±0.22

3.92±0.45

0.84

  Proteinuria g/L

2.07±0.40

28.33±10.35

0.02

3.42±0.44

0.76±0.31

0.0001

  Albuminuria mg/24

74.04±24.37

1.88±0.00

0.006

111.23±29.60

19.29±7.15

0.009

  Urinary creatinine   mg/kg/day

2.99±0.84

31.40±15.23

0.07

2.04±0.54

1.65±0.22

0.53

  Hematuria erythrocytes

40.54±18.61

10.80±0.00

0.12

35.00±12.92

31.43±20.80

0.88

  Glomerular filtration rate mL/min/1.73m2

80.60±7.36

93.49±13.88

0.42

92.84±10.60

87.65±14.20

0.77

Values are expressed as mean ± standard error (SE) and t-student test of dependent samples. t Class III vs. Class IV baseline comparation. Chi2 was used for dichotomous variables. CRP= C - reactive protein, C3 = complement protein C3, C4 = complement protein C4.

Comment. Longitudinal data for TLR7 and TLR9 are valuable. How about comparing the longitudinal changes in TLR7 and TLR9 between groups? How about comparing ΔTLR7 and 9 or using a linear mixed-effect model?

Answer. We made the comparison between the delta values ​​obtained from baseline and 12-month measurements. However, the longitudinal change did not show significant differences between the two groups. This table was not included in results. If we are required to include it in results, please let me know

Comparison between longitudinal changes in TLR7 and TLR9

CLASS III

CLASS IV

p

ΔTLR7

2.64 ± 1.10

0.71 ± 0.29

0.11

ΔTLR9

0.44 ± 0.17

0.17 ± 0.02

0.14

Comment. Is the title “Graph 1” appropriate for the authors’ instruction? I think “Figure” is better and more general.

Answer. Graph 1 was changed to Figure 2 

Comment. The analyses of the association (or correlation) between TLR7 (and 9) and clinical characteristics should be informative. What kind of parameters are associated with TLR7 or TLR9?

Answer. A correlation analysis was performed to find associations between TLR7 and 9 values ​​and other clinical parameters. No significant correlation coefficients were found associated with TLR7 and 9.

Comment. The authors must have the details of pathological findings (because all patients were categorized into LN Classes). So, please analyze the association between pathological findings and TLR7 (and TLR9).

Answer. A correlation analysis was performed to find associations between TLR7 and 9 values ​​and other clinical parameters. No significant correlation coefficients were found associated with TLR7 and 9.

Comment. The authors defined a significant difference as p≤ 0.05, but p<0.05 is more common.

Answer. p≤0.05 was modified to p<0.05 

Comment. Are the healthy controls in this study appropriate for the study?

Answer. We did not find information in the available scientific literature on the plasma expression of TLR 7 - TLR9 in patients with lupus nephritis. We consider that the results could be a little credible or confusing if we did not include a healthy control group to contrast

Comment. For example, did the authors compare between SLE patients without LN and patients with LN class III and IV? And/or between LN patients who achieved a complete remission and patients with LN class III and IV? This type of comparison should be helpful in drawing conclusions.

Answer. We only included patients with systemic lupus erythematosus who were recently sent to the Nephrologist with alterations in renal function. The patients underwent renal biopsy and the histopathological result reported Class III and Class IV.

According to our results, it is worth including patients in complete remission and without alterations in renal function in other study. 

Comment. For the flow chart, please state how many patients were enrolled and then how many were excluded for specific reasons. Also, "N-15" should be changed to, for example, "N=15".

Answer. N=15 patients from Class III, N=16 from Class IV, and N=17 from the healthy control group were enrolled. We modified the flowchart as requested. Three patients died before twelve months of follow-up. The results of those patients were excluded from the final analysis 

Figure 1

Comment. How did the authors select the healthy controls? Why was the n number of healthy controls 17?

Answer. Blood donors were selected because before becoming donors they go through multiple filters. Blood donors are considered predominantly healthy

The blood of 17 blood donors was included because they arrived on the day they were invited to donate an extra 10 mL of blood and all 17 agreed to participate in the study

Comment. For the methods, what is the “baseline”? please provide the definition.

Answer. When the Rheumatologist in charge of patients with systemic lupus erythematosus detects alterations in kidney function, he sends the patient for evaluation to the Nephrologist. The baseline determination was made at the first appointment with the Nephrologist for the blood sample and the kidney biopsy

Comment. For the data analysis of TLR7 and TLR9, were these values measured at the same plate (kit) on the same date? How does the distribution of the value of TLR7 and TLR9? Are these normal distributions without any standardization? I do not mean to criticize but I think these adjustments or quality control of the data may help the authors to find important results.

Answer. The TLR7 kit was run one day and the TLR9 kit was run the next day. In both kits, a standard curve was made to determine the concentrations of the absorbances. The analysis for both groups were made on the same plate and date. Concerning the distribution of values, we clarify the adjustments in the statistics section as shown below:

“Shapiro Wilk test was used to determine the distribution of values. For parametric comparisons, Paired-Sample T-Test and Independent Sample T-test tests were used. For non -parameter comparisons, Wilcoxon and Mann Whitney U tests were used”

Comment. If the authors want to mention the data at 24 months, information on treatment should be important. Please add these to the results.

Answer. SLE patients were treated with glucocorticoids, non-steroidal anti-inflammatory drugs, antimalarials, immunosuppressants, and biological products by the Department of Rheumatology

Comment. So please rethink the content and re-analyzIJe the data that the authors have.

Answer. Statistical data were re-analyzed. I hope the results are clearer

Comment. Then I think the authors can come up with a better conclusion and state the purpose of the study. (For example, low levels of TLR7 and TLR9 may be useful to diagnose LN class III or IV). This kind of conclusion should be useful for clinicians.

Answer. Low expression of TLR9 and increased TLR7 could be useful in the recent diagnosis of Class III and Class IV LN.

It is worth comparing the expression of both TLRs in LN with or without stable LN; with a larger number of patients and with a longer follow-up over time.

Reviewer 2 Report

Comments and Suggestions for Authors

Comments:

   The manuscript describes "The Expression of Toll-Like Receptors (TLR7 and TLR9) in Class III and Class IV of Recently Diagnosed Lupus Nephritis with Twelve Month Follow-up”. Renal involvement is an important cause of morbidity and mortality in systemic lupus erythematosus.  In this study, an ELISA method was used to determine the plasma expression of TLR7 and TLR9 proteins.  Expression of TLR9 was found to exhibit the opposite behavior to TLR7, with persistent hematuria and no change in glomerular filtration rate in all patients. Rheumatological data appeared to improve, but renal function data remained inconsistent, but several points need clarification.

Comment:

1. In Table 1, all abbreviations should be marked.

2. The overexpression of TLR7 can be observed in the baseline 159 determination and at twelve months of follow-up in LN Class III patients compared to the 160 expression found in HC, but not in the Class IX patients. The authors should discuss the results.

3. Graph 1. The author should include statistical comparisons of different groups.

4. TLR7 expression is higher in women than in men, and reducing TLR7 activity may slow the development of SLE. What is the difference in the expression of TLR7 or TLR9 in lupus nephritis class III and class IV among female patients in this study?

Comments on the Quality of English Language

Moderate editing of English language required

Author Response

Document TLR7 and TLR9

Reviewer

The manuscript describes "The Expression of Toll-Like Receptors (TLR7 and TLR9) in Class III and Class IV of Recently Diagnosed Lupus Nephritis with Twelve Month Follow-up”. Renal involvement is an important cause of morbidity and mortality in systemic lupus erythematosus.  In this study, an ELISA method was used to determine the plasma expression of TLR7 and TLR9 proteins.  Expression of TLR9 was found to exhibit the opposite behavior to TLR7, with persistent hematuria and no change in glomerular filtration rate in all patients. Rheumatological data appeared to improve, but renal function data remained inconsistent, but several points need clarification. 

    Comment. 1. In Table 1, all abbreviations should be marked.

   I appreciate your comments, we know that you intend for the document to be clear and improve the quality

   Answer. Values are expressed as mean ± standard error (SE).t-student test of dependent samples and Wilcoxon test were used. t Class III vs. Class IV baseline comparison. Chi2 was used for dichotomous variables. CRP= C - reactive protein, C3 = complement protein C3, C4 = complement protein C4.

   Comment. 2. The overexpression of TLR7 can be observed in the baseline 159 determination and at twelve months of follow-up in LN Class III patients compared to the 160 expression found in HC, but not in the Class IX patients. The authors should discuss the results.

   Answer. We were unable to find the recommended article in the available scientific literature. I would be very grateful if you could send me the bibliographical reference and I will gladly add it to the discussion of the document

   Comment. 3. Graph 1. The author should include statistical comparisons of different groups.

Answer. A correlation analysis was performed to find associations between TLR7 and 9 values ​​and other clinical parameters. No significant correlation coefficients were found associated with TLR7 and 9.

  Comment. 4. TLR7 expression is higher in women than in men, and reducing TLR7 activity may slow the development of SLE. What is the difference in the expression of TLR7 or TLR9 in lupus nephritis class III and class IV among female patients in this study?

  Answer. The table shows the results between women and men at baseline and twelve months follow-up for TLR7 and TLR9. No statistically significant difference was found. This table is not included in the main document.

Table. TLR7 and TLR9, Baseline vs. Twelve months follow-up  -  Female  -  Male

Class III Female

Class III Male

Baseline

Twelve months

p

Baseline

Twelve months

p

TLR7 ng/mL

2.73±1.21

0.90±0.27

0.15

1.38±1.16

1.49±1.16

0.95

TLR9 ng/mL

0.38±0.04

0.47±0.08

0.71

1.20±0.59

0.63±0.23

0.41

Class IV Female

Class IV

TLR7 ng/mL

0.53±0.14

0.63±0.23

0.71

0.62±0.54

1.83±1.53

0.43

TLR9 ng/mL

0.38±0.04

0.47±0.08

0.32

0.44±0.06

0.66±0.21

0.28

Reviewer 3 Report

Comments and Suggestions for Authors

The article is original and very interesting, but encadrable in Short Communication, in actual stage.

The topic is very relevant, the aim of this research being to demonstrate that, since LN represents the most serious organic manifestation of SLE in terms of morbidity and mortality, TLR modulation and signaling have become important strategies for the diagnosis and treatment.

Methodology is very good, author using the research findings on emerging molecular discoveries and approaches. Since the authors have mentioned The diagnosis of LN Class was determined by renal biopsy within the first three months of the detection of LN according to the criteria of the WHO or the International Society of Nephrology and the Renal Pathology Society, I recommend to include some suggestive histopathology and immunohistochemistry images of their original cases.

The conclusions are  very consistent with the evidence and arguments presented, the authors demonstrating a significant imbalance in TRL7 expression characterized by the increase in Class III and Class IV patients at baseline and twelve months of follow-up compared to the expression found in HC.

The references are very relevant, including also some relevant author’s previous experience in the field.

I suggest some supplementary corrections

1.     The Abstract should be more conciseded, to not repeat the results

2.     Introduction could be more detailed, including a definition of Lupus.

Page 6, Graph 1 could be Fig 1 and expressed in different colours, to be more attractive

Actual Fig 1 also could be better elaborated to be more attractive (see Biorender.com)

Comments on the Quality of English Language

The authors should follow carefully the Authors Guide for MDPI articles

Author Response

Document TLR7 and TLR9

Reviewer 3

Comments and Suggestions for Authors

The article is original and very interesting, but encadrable in Short Communication, in actual stage.

The topic is very relevant, the aim of this research being to demonstrate that, since LN represents the most serious organic manifestation of SLE in terms of morbidity and mortality, TLR modulation and signaling have become important strategies for the diagnosis and treatment.

Methodology is very good, author using the research findings on emerging molecular discoveries and approaches. Since the authors have mentioned The diagnosis of LN Class was determined by renal biopsy within the first three months of the detection of LN according to the criteria of the WHO or the International Society of Nephrology and the Renal Pathology Society,

Comment. I recommend including some suggestive histopathology and immunohistochemistry images of their original cases.

Answer. Voy a buscar mañana al Dr. Medina para ver si me consigue una demostrativa de la clase III y IV de HyE e inmunohistoquímica de reciente diagnóstico. La patóloga de Nefro está de vacaciones, parece que regresa mañana, el viernes el Dr. Medina tiene cita con ella y el viernes o el lunes me avisa si ya cuenta con las imágenes que tal vez sería de HyE e inmunofluorescencia. Comentan que inmunohistoquímica no hacen frecuentemente

The images are missing. The Pathologist is on vacation and returns on Wednesday, I hope to be able to get them when she returns

The conclusions are very consistent with the evidence and arguments presented, the authors demonstrating a significant imbalance in TRL7 expression characterized by the increase in Class III and Class IV patients at baseline and twelve months of follow-up compared to the expression found in HC.

The references are very relevant, including also some relevant author’s previous experience in the field.

I suggest some supplementary corrections

Comment. 1.     The Abstract should be more conciseded, to not repeat the results

Answer.

Abstract: Renal involvement is an important cause of morbidity and mortality in systemic lupus erythematosus (SLE). The present study included patients with recently diagnosed Class III and Class IV lupus nephritis (LN) treated by Rheumatology who, upon detecting alterations in kidney function, were referred to Nephrology for joint management of both medical specialties. The purpose of the study was to compare the plasma expression of TLR7 and TLR9 in healthy control (HC) subjects and newly diagnosed Class III and Class IV LN patients with twelve-month follow-ups. Plasma expression of TLR7 and TLR9 proteins was determined by the ELISA method. A significant increase in the expression of TLR7 protein was found in Class III LN in the basal determination compared to the expression in the HC (p=0.002) and at twelve months of follow-up (p=0.03) vs. HC. The expression of TLR9 showed the opposite behavior to TLR7 with a decrease in protein expression in the basal and final determination of LN Class III. The result was similar in the basal and final determination of LN Class IV compared to the expression in HC.

A significant decrease in SLEDAI -2K was observed at twelve months of follow-up in patients in Class III (p=0.01) and Class IV (p=0.0001) of LN. Complement C3 levels improved significantly at twelve-month follow-up in Class IV patients (p=0.0001). Complement C4 levels decreased significantly at twelve-month follow-up in LN Class III compared to baseline (p=0.01).

Anti-DNA antibodies decreased significantly at twelve months of follow-up in Class IV LN (p=0.01). A significant increase in proteinuria was found at twelve months of follow-up in Class III LN, compared to the baseline determination (p=0.02). In LN Class IV, proteinuria decreased at twelve months of follow-up compared to baseline (p=0.0001). Albuminuria decreased at twelve months of follow-up in LN Class IV (p=0.006). Also in Class IV LN, albuminuria decreased at twelve months of follow-up (p=0.009). Hematuria persisted in all patients and the glomerular filtration rate did not change. Three Class IV patients died before twelve months of follow-up from various causes.

In conclusion, although rheumatologic data appeared to improve, renal function data remained inconsistent. Low expression of TLR9 and increased TLR7 could be useful in the recent diagnosis of Class III and Class IV LN

Comment. 2.     The introduction could be more detailed, including a definition of Lupus.

Answer. The introduction was more detailed. I hope it is sufficiently defined for this document 

Comment.   Page 6, Graph 1 could be Fig 1 and expressed in different colors, to be more attractive

Answer. Figure 2 was modified as follows. I hope the colors are more acceptable

Figure 2

Reviewer 4 Report

Comments and Suggestions for Authors

The manuscript was submitted by José Ignacio Cerillos-Gutiérrez and Colab. This original paper can be important to further studies on establishing the role of Toll-like receptors (TLR7 and TLR9) in mediating systemic lupus erythematosus disease progression. They offer unusual information on the evolution of lupus nephritis accompanied by the monitoring of immune markers that are little known in their role in lupus nephritis disease (Toll-like receptors TLR7 and TLR9). The twelve-month prospective follow-up allowed us to identify the appearance of significant changes in clinical parameters based on the lupus nephritis classification (Class III and Class IV).
The authors used LN (not lupus nephritis) in the abstract – it should be corrected. On the other hand, in this subject, they explain TLR ( Toll-like receptors). This should be standardized. In addition, other editorial errors are visible. The authors should carefully check and correct the text. Moreover, graph 1 has poor quality.

Author Response

Reviewer 4

Document TLR7 and TLR9

Comments and Suggestions for Authors

The manuscript was submitted by José Ignacio Cerillos-Gutiérrez and Colab. This original paper can be important to further studies on establishing the role of Toll-like receptors (TLR7 and TLR9) in mediating systemic lupus erythematosus disease progression.

They offer unusual information on the evolution of lupus nephritis accompanied by the monitoring of immune markers that are little known in their role in lupus nephritis disease (Toll-like receptors TLR7 and TLR9).

The twelve-month prospective follow-up allowed us to identify the appearance of significant changes in clinical parameters based on the lupus nephritis classification (Class III and Class IV).

Comment. The authors used LN (not lupus nephritis)

Answer. I apologize for the lack of care in writing the document. Lupus Nephritis was included before LN

Comment. in the abstract – it should be corrected.

Answer. Lupus nephritis (LN) was corrected in the abstract

Comment. On the other hand, in this subject, they explain TLR (Toll-like receptors). This should be standardized.

Answer. The TLR7 and TLR9 ELISA method was standardized using the standard curve

Comment. In addition, other editorial errors are visible.

Answer. Other visible editorial errors have been corrected. I hope they have all been corrected

Comment. The authors should carefully check and correct the text.

Answer. We check the text and correct errors. I hope they have all been corrected

Comment. Moreover, graph 1 has poor quality.

Answer. Graph 1 was modified, it is now Figure 1, I hope the quality is better

Figure 1.

Round 2

Reviewer 1 Report

Comments and Suggestions for Authors

The authors addressed my concerns. But there are still a few points to add to the manuscripts. 

1. Please provide us with the results of the statistical relationship between TLR7 (as well as TLR9) and clinical parameters (Age, sex, GFR...) including pathological findings (active lesions like...cellular crescent and so on..) as Tables.  

Although these results can be placed as supplementary results, they are necessary for readers interested in TLR7 and TLR9.

2. Please provide us with the information on the treatment as the Table with statistical comparisons (it can be combined in Table 1).

Otherwise, we cannot specifically understand what kind of treatment was done on the patients in this study.

Author Response

Reviewer 1

Comments and Suggestions for Authors

The authors addressed my concerns. But there are still a few points to add to the manuscripts. 

Comment. 1. Please provide us with the results of the statistical relationship between TLR7 (as well as TLR9) and clinical parameters (Age, sex, GFR...) including pathological findings (active lesions like...cellular crescent and so on..) as Tables. Although these results can be placed as supplementary results, they are necessary for readers interested in TLR7 and TLR9.

Answer. We performed a non-parametric correlation test using the delta values ​​of TRL7, TRL9 and other clinical variables related to kidney function and rheumatological data. The class III group presented a negative correlation between the change of TRL9 and urea and creatinine. For Class III patients, increasing change in TRL9 levels over time is associated with less change in renal function represented by urea and creatine. On the other hand, for Class IV patients, changes in TRL9 are associated with a change in long-term GFR and magnesium. However, the comparison between the baseline values ​​and those at 12 months in Table 1 did not show differences for both classes, for this reason we cannot specify whether these delta correlations are due to the increase or decrease in the TRL9 value.

Correlation between delta values of TRL7 and TRL9 with clinical parameters of renal function and rheumatological data

Class III

Class IV

ΔTLR7

ΔTLR9

ΔTLR7

ΔTLR9

Rho

p

Rho

p

Rho

p

Rho

p

ΔSLEDAI -2K

-0.595

0.159

0.739

0.058

0.224

0.533

-0.576

0.082

ΔC3

-0.262

0.531

-0.119

0.779

-0.100

0.770

-0.345

0.328

ΔC4

-0.071

0.867

0.167

0.693

0.336

0.312

-0.115

0.751

ΔCRP

-0.100

0.873

-0.800

0.104

-0.261

0.467

-0.224

0.533

ΔUrea

0.183

0.637

-0.667

0.050

0.227

0.502

0.176

0.627

ΔCreatinine

0.209

0.589

-0.703

0.035

0.018

0.960

0.559

0.093

ΔGFR

0.283

0.460

-0.467

0.205

0.006

0.987

0.661

0.038

ΔProteinuria

0.427

0.252

-0.435

0.242

-0.450

0.224

0.567

0.112

ΔUrinary creatinine  

-0.800

0.200

-0.800

0.200

-0.857

0.014

0.321

0.482

ΔHematuria

0.400

0.600

-0.400

0.600

0.357

0.432

0.214

0.645

ΔPhosphorus

0.429

0.337

-0.036

0.939

0.091

0.790

0.401

0.250

ΔCalcium

0.393

0.383

-0.321

0.482

-0.336

0.312

0.394

0.260

ΔMagnesium

0.516

0.295

-0.577

0.231

-0.213

0.529

0.665

0.036

Delta values ​​were obtained from the absolute difference between baseline and 12-month values. Spearman's Rho correlation test was used with the delta values. GFR=Glomerular filtration rate, CRP= C - reactive protein, C3 = complement protein C3, C4 = complement protein C4.

I would be very grateful if you could inform me if it is necessary to include this information in the main document.

Comment. 2. Please provide us with the information on the treatment as the Table with statistical comparisons (it can be combined in Table 1).

Answer: Table 1 was modified to include pharmacological treatments for patients with LN Class III and Class IV

Table 1. Class III and Class IV Lupus Nephritis; Baseline vs. Twelve Months Follow-up

Class III

N=15

      Class IV

          N=16                   

Baseline

Twelve months

p

Baseline

Twelve months

p

  Gender n (%)

     Male

3 (20)

3 (18.75)

0.93t

     Female

12 (80)

13 (81.25)

  Age years

28.12±1.75

31.56±1.82

0.18 t

  Body weight Kg

67.35±5.47

63.97±3.66

0.61 t

  Body mass index  kg/m2

 24.95±1.26

25.66±1.47

0.72 t

Rituximab

0.92

No treatment

11 (73)

12 (75)

treatment

4 (27)

4 (25)

Tacrolimus

0.16

No treatment

15 (100)

14 (88)

treatment

0 (0)

2 (13)

Cyclosporine

0.025 t

No treatment

9 (60)

15 (94)

treatment

6 (40)

1 (6)

Mycophenolate mofetil

0.89

No treatment

4 (27)

6 (38)

treatment

11 (73)

10 (62)

Methylprednisolone

0.52

No treatment

6 (40)

6 (38)

treatment

9 (60)

10 (62)

Prednisone

0.32

No treatment

4 (27)

2 (13)

treatment

11 (73)

14 (87)

Azathioprine

No treatment

14 (93)

15 (94)

0.96

treatment

1 (7)

1 (6)

Chloroquine

0.17

No treatment

8 (53)

6 (38)

treatment

7 (47)

10 (62)

Biochemical data

  Hemoglobin g/dL

12.15±0.57

12.37±0.76

0.82

11.46±0.56

12.47±0.89

0.33

  Hematocrit %

35.57±1.52

37.72±2.66

0.49

33.99±1.56

36.85±1.54

0.21

  Platelets thousands/µL

239.33±19.22

276.45±19.68

0.19

231.44±22.84

293.73±19.82

0.05

  Glucose mg/dL

89.93±2.52

84.89±3.66

0.27

96.64±3.00

91.08±4.05

0.27

  Albumin g/dL

3.58±0.23

3.46±0.27

0.74

3.37±0.17

3.46±0.27

0.77

  Chlorine mmol/L

109.75±1.60

104.43±2.14

0.05

107.35±1.12

109.18±3.49

0.60

  Potassium mmol/L

4.35±0.24

4.26±0.25

0.80

4.54±0.21

4.43±0.14

0.68

  Calcium mmol/L

3.09±0.58

4.72±0.89

0.14

2.93±0.47

4.74±1.02

0.10

  Magnesium mmol/L

0.92±0.10

1.03±0.18

0.60

0.92±0.07

1.29±0.20

0.08

  Sodium mmol/L

137.47±0.76

136.96±0.70

0.76

138.15±0.82

139.09±0.78

0.60

Rheumatological Data

  SLEDAI -2K

18.53±1.87

9.88±2.61

0.01

23.31±1.63

10.80±1.74

0.0001

  C3 mg/dL

107.51±9.84

91.78±12.75

0.13

65.01±4.50

104.68±6.85

0.0001

  C4 mg/dL

23.19±1.87

14.98±2.42

0.01

19.52±3.57

20.45±2.64

0.84

  CRP mg/L

7.60±1.64

6.49±2.24

0.69

14.64±1.63

9.83±2.50

0.11

  Ab anti-DNA IU/mL

111.49±58.71

( - )

0.07

72.33±17.04

22.36±4.51

0.01

Renal Function

  Urea mg/dL

34.73±2.22

50.43±12.61

0.23

47.23±5.31

46.21±11.95

0.94

  Creatinine mmol/L

1.11±0.11

1.89±0.78

0.33

1.09±0.17

1.65±0.74

0.44

  Phosphorus mmol/L

4.11±0.22

4.00±0.42

0.81

3.82±0.22

3.92±0.45

0.84

  Proteinuria g/L

2.07±0.40

28.33±10.35

0.02

3.42±0.44

0.76±0.31

0.0001

  Albuminuria mg/24

74.04±24.37

1.88±0.00

0.006

111.23±29.60

19.29±7.15

0.009

  Urinary creatinine   mg/kg/day

2.99±0.84

31.40±15.23

0.07

2.04±0.54

1.65±0.22

0.53

  Hematuria erythrocytes

40.54±18.61

10.80±0.00

0.12

35.00±12.92

31.43±20.80

0.88

  Glomerular filtration rate mL/min/1.73m2

80.60±7.36

93.49±13.88

0.42

92.84±10.60

87.65±14.20

0.77

I would be very grateful if you could inform me if it is necessary to include this information in the main document.

Comment. Otherwise, we cannot specifically understand what kind of treatment was done on the patients in this study.

Exposure for this cohort is based on the diagnosis of lupus nephritis

Reviewer 2 Report

Comments and Suggestions for Authors

Accepted

Comments on the Quality of English Language

Minor editing of English language required

Author Response

Thank you very much for your comments, they were very useful in improving the document

Reviewer 3 Report

Comments and Suggestions for Authors

The authors have made all the corrections suggested. I recommend the acceptance of the article in this revised form

Author Response

(The authors gave the same response as above.)

Round 3

Reviewer 1 Report

Comments and Suggestions for Authors

The authors partially addressed my concerns.

For the correlation between TLR7,9 and clinical valuables, besides this analysis regarding delta, we need the cross-sectional analyses at the baseline. This can indicate that what kind of factors potentially influence to the concentration of TLR7, 9. This should be the important information for the novel study involving in TLR7, 9. Even if the correlation between them are all negative, we need this for this study. In addition, don't authors have the detailed pathological information about these patients? please provide the information about cross-sectional association between TLR7,9 at baseline and pathological findings at the baseline. This may indicate deeper insight of TLR7,9 in LN patients. That is why the reviewe sticks to this point.  Please put these results in the manuscript.

(Or, have the authors already mentioned this in supplementary tables? In that case, somehow, I cannot see supplementary materials via this reviewing system...) 

For the point 2, revised Table 1 is fine. Just one point, we do not need both "treatment" and "no treatment". we can understand each other if the author provide us one. So, please provide us only N and percentage of "treatment" and put this revised Table1 in the manuscript.

Author Response

Reviewer 3

Comments and Suggestions for Authors

The authors partially addressed my concerns. 

Comment. For the correlation between TLR7,9 and clinical valuables, besides this analysis regarding delta, we need the cross-sectional analyses at the baseline. This can indicate that what kind of factors potentially influence to the concentration of TLR7, 9. This should be the important information for the novel study involving in TLR7, 9. Even if the correlation between them are all negative, we need this for this study. In addition, don't authors have the detailed pathological information about these patients? please provide the information about cross-sectional association between TLR7,9 at baseline and pathological findings at the baseline. This may indicate deeper insight of TLR7,9 in LN patients. That is why the reviewe sticks to this point.  Please put these results in the manuscript.

Answer. A cross-sectional correlation was performed to determine the association between clinical parameters related to renal function and rheumatological data at baseline. A positive correlation was found between TLR7 and CRP values ​​for NL class IV patients. These results are shown in Table 3 which is included in the manuscript.

Table 3. Cross-sectional correlation between basal TLR7 and TLR9 with clinical parameters of renal function and rheumatological data

Class III

N=15

Class IV

N=16

TLR7

TLR9

TLR7

TLR9

Rho

p

Rho

p

Rho

p

Rho

p

SLEDAI -2K

0.116

0.71

-0.061

0.84

0.171

0.53

0.317

0.23

C3

0.104

0.73

-0.187

0.54

-0.395

0.13

0.110

0.68

C4

-0.005

0.99

-0.300

0.32

-0.469

0.07

-0.147

0.59

CRP

-0.281

0.35

0.209

0.49

0.540

0.031

-0.409

0.12

Urea

0.093

0.76

0.319

0.29

0.053

0.85

0.077

0.79

Creatinine

-0.011

0.97

-0.319

0.29

-0.145

0.59

0.024

0.93

GFR

-0.069

0.82

0.354

0.24

-0.140

0.62

0.159

0.57

Proteinuria

0.005

0.99

0.374

0.21

0.115

0.67

-0.056

0.84

Urinary creatinine

-0.295

0.33

0.148

0.63

-0.044

0.87

0.141

0.60

Hematuria

-0.173

0.61

-0.290

0.39

-0.189

0.56

0.315

0.32

Phosphorus

-0.180

0.58

0.322

0.31

0.178

0.51

-0.160

0.55

Calcium

0.198

0.52

0.281

0.35

0.028

0.92

-0.130

0.63

Magnesium

-0.274

0.37

0.238

0.43

-0.137

0.61

0.189

0.48

Spearman's Rho correlation test. GFR=Glomerular filtration rate, CRP= C - reactive protein, C3 = complement protein C3, C4 = complement protein C4.

Comment. (Or, have the authors already mentioned this in supplementary tables? In that case, somehow, I cannot see supplementary materials via this reviewing system...) 

Answer. The above results were not included in supplementary tables. However, we added the new suggested results to the manuscript. 

Comment. For the point 2, revised Table 1 is fine. Just one point, we do not need both "treatment" and "no treatment". we can understand each other if the author provide us one. So, please provide us only N and percentage of "treatment" and put this revised Table1 in the manuscript.

Answer: Table 1 was modified to improve the description of the percentages of n of the treatments as shown below: 

Table 1. Class III and Class IV Lupus Nephritis; Baseline vs. Twelve Months Follow-up

Class III

N=15

      Class IV

          N=16                   

Baseline

Twelve months

p

Baseline

Twelve months

p

  Gender n (%)

     Male

3 (20)

3 (18.75)

0.93t

     Female

12 (80)

13 (81.25)

  Age years

28.12±1.75

31.56±1.82

0.18 t

  Body weight Kg

67.35±5.47

63.97±3.66

0.61 t

  Body mass index  kg/m2

 24.95±1.26

25.66±1.47

0.72 t

Treatment

Rituximab

4 (27)

4 (25)

0.92

Tacrolimus

0 (0)

2 (13)

0.16

Cyclosporine

6 (40)

1 (6)

0.025 t

Mycophenolate mofetil

11 (73)

10 (62)

0.89

Methylprednisolone

9 (60)

10 (62)

0.52

Prednisone

11 (73)

14 (87)

0.32

Azathioprine

1 (7)

1 (6)

Chloroquine

7 (47)

10 (62)

0.17

Biochemical data

  Hemoglobin g/dL

12.15±0.57

12.37±0.76

0.82

11.46±0.56

12.47±0.89

0.33

  Hematocrit %

35.57±1.52

37.72±2.66

0.49

33.99±1.56

36.85±1.54

0.21

  Platelets thousands/µL

239.33±19.22

276.45±19.68

0.19

231.44±22.84

293.73±19.82

0.05

  Glucose mg/dL

89.93±2.52

84.89±3.66

0.27

96.64±3.00

91.08±4.05

0.27

  Albumin g/dL

3.58±0.23

3.46±0.27

0.74

3.37±0.17

3.46±0.27

0.77

  Chlorine mmol/L

109.75±1.60

104.43±2.14

0.05

107.35±1.12

109.18±3.49

0.60

  Potassium mmol/L

4.35±0.24

4.26±0.25

0.80

4.54±0.21

4.43±0.14

0.68

  Calcium mmol/L

3.09±0.58

4.72±0.89

0.14

2.93±0.47

4.74±1.02

0.10

  Magnesium mmol/L

0.92±0.10

1.03±0.18

0.60

0.92±0.07

1.29±0.20

0.08

  Sodium mmol/L

137.47±0.76

136.96±0.70

0.76

138.15±0.82

139.09±0.78

0.60

Rheumatological Data

  SLEDAI -2K

18.53±1.87

9.88±2.61

0.01

23.31±1.63

10.80±1.74

0.0001

  C3 mg/dL

107.51±9.84

91.78±12.75

0.13

65.01±4.50

104.68±6.85

0.0001

  C4 mg/dL

23.19±1.87

14.98±2.42

0.01

19.52±3.57

20.45±2.64

0.84

  CRP mg/L

7.60±1.64

6.49±2.24

0.69

14.64±1.63

9.83±2.50

0.11

  Ab anti-DNA IU/mL

111.49±58.71

( - )

0.07

72.33±17.04

22.36±4.51

0.01

Renal Function

  Urea mg/dL

34.73±2.22

50.43±12.61

0.23

47.23±5.31

46.21±11.95

0.94

  Creatinine mmol/L

1.11±0.11

1.89±0.78

0.33

1.09±0.17

1.65±0.74

0.44

  Phosphorus mmol/L

4.11±0.22

4.00±0.42

0.81

3.82±0.22

3.92±0.45

0.84

  Proteinuria g/L

2.07±0.40

28.33±10.35

0.02

3.42±0.44

0.76±0.31

0.0001

  Albuminuria mg/24

74.04±24.37

1.88±0.00

0.006

111.23±29.60

19.29±7.15

0.009

  Urinary creatinine   mg/kg/day

2.99±0.84

31.40±15.23

0.07

2.04±0.54

1.65±0.22

0.53

  Hematuria erythrocytes

40.54±18.61

10.80±0.00

0.12

35.00±12.92

31.43±20.80

0.88

  Glomerular filtration rate mL/min/1.73m2

80.60±7.36

93.49±13.88

0.42

92.84±10.60

87.65±14.20

0.77

Round 4

Reviewer 1 Report

Comments and Suggestions for Authors

The authors addressed my concerns.